# Protocol for a scoping review of multi-omic analysis for rare diseases

Katie Kerr, Helen McAneney, Amy Jayne McKnight

## ABSTRACT

**Introduction** The development of next generation sequencing technology has enabled cost-efficient, large scale, multiple 'omic' analysis, including epigenomic, genomic, metabolomic, phenomic, proteomic and transcriptomic research. These integrated approaches hold significant promise for rare disease research, with the potential to aid biomarker discovery, improve our understanding of disease pathogenesis and identify novel therapeutic targets. In this paper we outline a systematic approach for a scoping review designed to evaluate what primary research has been performed to date on multi-omics and rare disease.

**Methods and analysis** This protocol was designed using the Joanna Briggs Institute methodology for scoping reviews. Databases to be searched will include: MEDLINE, EMBASE, PubMed, Web of Science, Scopus and Google Scholar for primary studies relevant to the key terms 'multi-omics' and 'rare disease', published prior to 30th December 2018. Grey literature databases GreyLit and OpenGrey will also be searched, as well as reverse citation screening of relevant articles and forward citation searching using Web of Science Cited Reference Search Tool. Data extraction will be performed using customised forms and a narrative synthesis of the results will be presented.

**Ethics and dissemination** As a secondary analysis study with no primary data generated, this scoping review does not require ethical approval. We anticipate this review will highlight a gap in rare disease research and provide direction for novel research. The completed review will be submitted for publication in peer-reviewed journals and presented at relevant conferences discussing rare disease research and/or molecular strategies.

## Strengths and limitations of this study

► The proposed scoping review will evaluate primary research that has been conducted using multi-omic approaches for rare disease.
► The data extracted will include the type of 'omic' research, study design, methodological rigour of the studies and outcomes of interest.
► As it would be impractical to attempt to list every rare disease as an individual search term, this review will be limited to research which explicitly specifies in the article that the disease of focus is a rare disease.
► Rare diseases named in this approach will be checked for prevalence to ensure they meet the European Union definition of a rare disease.

## INTRODUCTION

Individual rare diseases are defined as affecting less than 5 in 10 000 people in Europe and less than 200 000 people in America at any given time (7.5 in 10 000).[1 2] There are estimated to be between 5000 and 8000 different types of rare diseases and cumulatively these affect approximately 350 000 000 people worldwide.[2] Due to the clinical heterogeneity of rare diseases, diagnosis is often difficult and treatment options limited. Rare conditions are often severely debilitating and/or life limiting, with 30% of patients dying before they reach their fifth birthday.[3] Furthermore, approximately 80% of rare conditions have a genetic cause, such as those resulting from a single mutation, polygenic inheritance or structural chromosome abnormalities.[4]

The development of massively parallel sequencing technologies in the past decade, compared with traditional sequencing methods such as Sanger sequencing, has improved the accuracy and speed of rare disease diagnosis.[5] Much of this next generation sequencing research has focused on genomic analysis, such as the 100 000 Genomes Project; a national project based in the UK attempting to sequence the genomes of people with rare diseases (including some cancers) and their family members. There is a need to build on these genomic resources by utilising integrated multi-omic analysis, as is increasingly employed for common complex diseases,[6] a biological approach to research that takes into consideration multiple omic terms, of which there are over 500.[7] These include: epigenomic, genomic, metabolomic, phenomic, proteomic and transcriptomic research.

While the majority of rare disease studies have improved diagnosis through genomic analysis, other omic strategies such as epigenomics and transcriptomics have been employed. In recent studies, differential

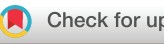

Centre for Public Health, Queen's University Belfast, Belfast, Northern Ireland, UK

**Correspondence to**
Dr Amy Jayne McKnight;
a.j.mcknight@qub.ac.uk

methylation has been identified as a potential marker of rare renal diseases such as IgA nephropathy and Autosomal dominant polycystic kidney disease[8 9] and rare ophthalmic diseases such as retinitis pigmentosa and Fuchs endothelial corneal dystrophy.[10 11] Additionally, transcriptomic analysis of retinal disease may aid in diagnosis and identifying new treatments.[12] Combining singular omic analysis with other analyses to form an allied multi-omic approach could provide new insights into rare disease molecular profiles, aid in diagnosis, prognosis prediction and identifying novel treatment targets. Following the publication of these singular omic analyses across a variety of multi-omic types, several recent review articles discuss this untapped potential. For example, the potential benefits of integrative analysis for rare neurological diseases,[13] dermatomyositis/polymyositis,[14] liver cancer[15] and inborn errors of metabolism.[16] This review will identify studies were more than one omic term has been employed in studies of rare disease.

A major challenge facing researchers is interpretation of the large amounts of data generated from next generation approaches. In contrast to common complex diseases, rare disease researchers are also faced with the challenge of generating reliable, high quality data, such as a randomised controlled trial, when they often have an extremely small sample size. To tackle this, international collaboration of research is crucial to include participants of a rare disease who may live in different countries, or even continents. Additionally, there is a need to develop a standardised reporting structure for rare disease omic analysis, such as that which exists for genome wide association studies.[17]

This scoping review will highlight important gaps in multi-omic rare disease research and will summarise elements of good practice for rare disease omic analyses, providing evidence for more robust, larger-scale collaborative studies.

### Review aim and objectives

This scoping review will aim to systematically summarise what research has been conducted into multi-omics and rare disease by:

▶ Evaluating what primary research studies exist pertaining to multi-omics and rare disease and which type of omic analysis was undertaken.
▶ Highlighting research outcomes with implications for rare disease diagnosis, treatment or improved understanding of disease mechanisms.

### METHODS

This scoping review protocol was designed using the Joanna Briggs Institute Reviewers Manual 2015—methodology for scoping reviews,[18] and our completed review will follow the Preferred Reporting Items for Systematic Reviews and Meta-Analyses Extension for Scoping Reviews guidelines.[19]

### Eligibility criteria

The inclusion criteria were developed using the population, concept and context (PCC) framework of the Joanna Briggs Institute scoping review methodology, with a deliberately wide scope to capture as many relevant research studies as possible. The 'population' of this scoping review will be rare diseases meeting the European definition of less than 1 in 2000.[2] Participants may be rare disease patients of all ages, ethnicity and gender. The 'concept' of the review is multi-omics and therefore must include at least two types of omic analysis. The most common omic terms we would expect to see are epigenomics, genomics, metabolomics, phenomics, proteomics and transcriptomics, however any omic term may be included. A comprehensive list of omic terms is available for reference on omics website.[7] The review 'context' will be within a primary research study written in English and published prior to the 30th December 2018 and as such other reviews of the literature will not be included in the final synthesis but may be helpful for the identification of further relevant studies through reverse screening and forward citation searching.

### Information sources and search terms

Articles will be identified primarily by searching the electronic databases MEDLINE via Ovid, EMBASE via Ovid, PubMed, Web of Science, Scopus and Google Scholar. This large selection of databases will be included in the strategy to perform as wide a search as possible. Online supplementary table S1 and S2 of the supplementary files illustrate example search terms for MEDLINE and Embase, which will be adapted for the other databases. As our search terms are limited to papers which directly define the condition as a 'rare disease', we have included Google Scholar to enable 'full text searching', in order to help address this limitation of our review. These terms were devised based on the PCC framework discussed above and discussed with a faculty librarian. In addition to traditional searching, reverse citation screening of the reference lists of relevant articles (ie, including the key terms such as multi-omics and rare disease) and forward citations (articles which have cited the identified papers) will be searched for using the Web of Science Cited Reference Search tool.

### Selection of studies, data extraction and analysis

The screening process to select studies will begin with compilation of articles identified from the several databases using reference management software (EndNote X8) and subsequent auto-removal of duplicates followed by additional manual checking and removal of further duplicates missed by the software. Title, keyword and abstract screening will identify any studies which noticeably do not meet the study inclusion criteria, that is, no discussion of multi-omics or singularly focusing on a common disease. Following this, any papers which initially appear to meet the inclusion criteria will be screened *via* reading the full text of the article. Data will then

be extracted from the articles which are selected to be included in the review using a customised data extraction form, (online supplementary table S3), to include information about study design, participant information, experimental methods and statistical analysis. Variables which may impact omic analysis will be highlighted and discussed where known, including participant age, co-morbidities, gender, ethnicity and type of sample collected. This information will be summarised in a table. The screening and data extraction will be performed in duplicate by two independent personnel to ensure reproducibility and any discrepancies will be considered by third reviewer. A narrative synthesis of the results of this review will be presented.

## Patient and public involvement

There was no public or patient involvement in the making of this protocol and will be none in the review, as we are using secondary data from publicly available resources. However, this review will form part of a wider patient-centred research strategy; a multi-omic approach to improving rare disease diagnosis within the *100 000 Genomes Project*, as a part of the Northern Ireland Genomic Medicine Centre and other genomic medicines centres across the UK. Collaborative patient-researcher conferences highlighted the need to prioritise improving diagnostic yield, such as the *'All-Ireland Rare Disease Day Conference'* in March 2018 and an '*Advancing registry and information resources for rare diseases'* event in January 2018.

## ETHICS AND DISSEMINATION

As this is a review of previously published data and no primary data will be collected, formal ethical committee approval is not required for this scoping review. This scoping review will highlight primary studies that have been conducted for multi-omic analysis of rare disease and discuss the quality of any such research. Multi-omic analysis of rare diseases has important implications for diagnostic/prognostic biomarker development, identifying new treatments and improving our understanding of the diseases themselves. Subsequently, any gaps in the literature which are identified will be presented at conferences, research group meetings and submitted for publication in a peer reviewed journal. This review will also form part of a chapter in KK's PhD thesis. This review may help guide future rare disease research.

**Acknowledgements** The authors would like to thank Richard Fallis, the School of Medicine, Dentistry and Biomedical Sciences Faculty librarian, for his consultation when developing the search terms.

**Contributors** AJMcK conceptualised the project, all authors researched the review protocol, KK initially drafted this review protocol, and all authors were involved in preparing and agreeing the final protocol manuscript.

**Funding** KK is supported by a Department for the Economy Co-operative Awards in Science and Technology (DfE-CAST) studentship award. Funding support was provided by the Medical Research Council – Northern Ireland Executive support of the Northern Ireland Genomic Medicine Centre though Belfast Health and Social Care Trust.

**Competing interests** None declared.

**Patient consent for publication** Not required.

**Provenance and peer review** Not commissioned; externally peer reviewed.

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
