## [Reviewer comments · BMJ Open]

ARTICLE DETAILS

TITLE (PROVISIONAL)	Protocol for a scoping review of multi-omic analysis for rare diseases.
AUTHORS	Kerr, Katie; McAneney, Helen; McKnight, Amy

VERSION 1 - REVIEW

REVIEWER	Gareth Baynam Undiagnosed Diseases Program, Genetic Services of Western Australia, Australia
REVIEW RETURNED	09-Oct-2018

GENERAL COMMENTS	This is a well written manuscript and a very timely and important protocol for review of the topic area
---

REVIEWER	Dr Clara van Karnebeek Amsterdam University Medical Centres, Amsterdam The Netherlands
REVIEW RETURNED	21-Oct-2018

GENERAL COMMENTS	The authors have formulated an ambitious aim, ie to perform and report a literature review on omics- studies in rare diseases. Their motivation is well described, including tackling the challenge of big data interpretation, small patient numbers inherent to rare diseases, and the need for standardized analysis and reporting of omics analysis/results. It is not completely clear to the reviewer how this systematic review will help address those challenges. 1) Can the reviewers clarify this, ie how will the results of this systematic review improve omics analysis in rare diseases?2) In diagnostic omics studies, usually the underlying diagnosis is unclear and a rare disease is expected base don phenotype but it is unsure whetther a rare diseasae is actually present. How do the authors expect to deaal with this?3) Have literature reviews on multi omics studies in rare diseases been performed and published? I believe this is the case for genomics and metabolomics, and for genomics and transcriptomics. The authors should list in the current manuscript the reviews that have been performed to date.4) Who do the authors include diagnostic, biomarker, and treatment studies in this review? Casting the net too wide could make the review too diffuse and difficult to interpret. Please address this concern.
--

	5) Most interesting for the reader would be a take away message from the literature review such as 'which multi-omics study designs are most successful? what are predictors of success? what are the do's and the don'ts?' Perhaps the authors can incorporate this into their review design. 6) The technology both for generating data and for interpreting data (bio informatics tools and resources) are developing at a high speed. Is it really faair to compare studies performed in 2012 to those in 2018? 7) Phenotyping and the extent to which this has been done and reported in the studies, will be central to success both in terms of diagnostic yield and discovery of diseases, biomarkers, treatments. How will the authors score and describe phenotyping? 8) Ethnicity of patients studied, sample types etc will also play a role in omics study yield; authors should list such variables in this manuscript and address inherent challenges.
--	---

REVIEWER	Fowzan S Alkuraya KFSHRC Saudi Arabia
REVIEW RETURNED	23-Oct-2018

GENERAL COMMENTS	Kerr and colleagues outline their plan for a scoping review on all multiomics studies involving rare diseases. The goal of the review is to highlight the power of multiomics in understanding these rare diseases. The review is likely to achieve this goal based on what the authors plan to abstract from those studies. I have the following suggestions: 1- As the authors acknowledge as a limitation, their review methodology will miss multiomics studies that do not explicitly use the term "rare" in the title, abstract or key words. The potential misses as a result is likely substantial. May I suggest to use tools that search the full text such as Google Scholar? I understand that this comes at a cost since they will likely have to sort through many more studies that may list the word "rare" without being specifically focused on rare diseases. 2- One minor error: "two examples of which for are available", please remove "for".
---

REVIEWER	Rosie Hanneke, Assistant Professor & Information Services/Liaison Librarian Library of the Health Sciences University of Illinois at Chicago Chicago, IL United States
REVIEW RETURNED	26-Nov-2018

GENERAL COMMENTS	1. Please indicate that your completed review will follow scoping review reporting guidelines from PRISMA. PRISMA-ScR extension: http://annals.org/aim/fullarticle/2700389/prisma-extension-scoping-reviews-prisma-scr-checklist-explanation 2. I would recommend that your team consult a librarian or other search expert for advice on your search strategy, including which databases to search and which terms to include.
---

	3. It is unclear which platforms you will use to search each database. Please indicate this in your Methods--e.g., MEDLINE (Ovid), EMBASE (Embase.com), etc. There is no need to search MEDLINE via both Ovid and PubMed. 4. The Cochrane Database of Systematic Reviews does not identify primary studies so I'm not sure why you would include this in your search. Cochrane Reviews are indexed for MEDLINE so you would already be finding these through that database if you do want to include them as means of identifying primary studies. 5. It is unnecessary to conduct the search twice. Have your search strategy reviewed by a librarian/search expert to ensure that the syntax is correct and reproducible and then run it once. On the other hand, you do need two individuals to screen articles, which you do not mention. Please indicate that two individuals will screen results, with a third resolving discrepancies. 6. It is not clear whether your search strategy as presented is broad enough to capture all relevant literature. I understand that there exist thousands of rare diseases and so it would likely not be feasible to search for all of these by name, but it seems that you will miss a large portion of the literature by limiting to only those articles with "rare disease" or one of its variants in the title, abstract or controlled vocabulary. Please explore whether it would be possible to include individual disease names in your search, even from an abbreviated list, in order to improve recall in your search--again, a librarian or other search expert could help you with this. Likewise "multiomics"--is there a way to identify articles that mention two individual omic terms in the abstract but do not use the specific term "multiomic"? If not possible, it is understandable but this should be explained as a limitation of your methodology. 7. Scoping reviews do not include critical appraisal of methodological quality -- see p. 17 of the JBI manual that you reference. This portion of your analysis should be removed. Please consult seminal scoping review methods papers (e.g. Arksey & O'Malley 2005, Levac, Colquhoun & O'Brien 2010) in addition to JBI to ensure that you are following best practices. Consulting experts about initial findings would be a useful step to add to your method.
--	---

REVIEWER	Erik Cobo Valeri Barcelona-Tech, Spain
REVIEW RETURNED	01-Dec-2018

GENERAL COMMENTS	This paper presents the protocol of an open review without the objective to test a confirmatory hypothesis. So there is no risk of selective outcome reporting and I think it may be published in its actual wording. Just a minor comment. Please consider if you should also pay attention to individualised trials (i.e., different interventions in repeated periods for stable, chronic conditions, such as "n-of-1" trials).
--

VERSION 1 – AUTHOR RESPONSE

Reviewer 1		
This is a well written manuscript and a very timely and important protocol for review of the topic area	We thank the reviewer for highlighting this timely and important protocol.	None.
Reviewer 2		
The authors have formulated an ambitious aim, i.e. to perform and report a literature review on omics- studies in rare diseases. Their motivation is well described, including tackling the challenge of big data interpretation, small patient numbers inherent to rare diseases, and the need for standardized analysis and reporting of omics analysis/results. It is not completely clear to the reviewer how this systematic review will help address those challenges. 1) Can the reviewers clarify this, ie how will the results of this systematic review improve omics analysis in rare diseases?	We thank the reviewer for the detailed consideration and feedback returned. This systematic review will provide the basis for multiomic research in a number of ways.  • First, we expect that this scoping review will highlight that there is a significant gap in multiomic rare disease research literature, demonstrating the need for further studies with harmonised phenotyping, consistent quality control, and robust analytical workflows, but a solid evidence base for these issues across multi rare diseases does not presently exist. This research may identify particular approaches for rare diseases that generate convincing outcomes and thus should be shared for larger-scale, collaborative studies. We plan to summarise diagnostic, prognostic and / or treatment outcomes. • Second, for any relevant studies returned, we intend to compare how these studies have addressed issues commonly associated with multiomic analyses; including big data interpretation and small patient numbers. In previous systematic reviews of epigenetics in rare renal and ophthalmic diseases (Kerr et al., submitted) sub-optimal 	We have clarified this in the main introduction text of manuscript. (Lines 110 – 112)

	methodological rigour was apparent. This review will form a more comprehensive analysis of the field, providing strong evidence of limitations and good practice for published multi-omic research for rare disease to date. This carefully performed review may reinforce the growing body of literature which calls for a collaborative approach to rare disease studies to improve participant numbers.	
2) In diagnostic omics studies, usually the underlying diagnosis is unclear and a rare disease is expected based on phenotype but it is unsure whether a rare disease is actually present. How do the authors expect to deal with this?	Although the diagnosis may not be known at the start of a diagnostic omic study, by the time the research is published there is usually a diagnosis or at least a clinical-molecular phenotype provided and should therefore be captured by our search strategy. There are several examples of an unclear underlying diagnosis that was subsequently identified and published, for example whole exome sequencing used to provide a molecular diagnosis for patients with Joubert Syndrome and other related syndromes (PMID: 23034536), targeted genome sequencing as part of the 100,000 Genomes Project and the UK Inherited Retinal Disease Consortium, which identified a rare and previous uncharacterised mutation in SRD5A3 thought to contribute to early onset retinal dystrophy (PMID: 28253385), and rare missense variants in PRPS1 leading to X-linked inheritance of retinal dystrophy in females discovered collaboratively by the 100,000 Genomes Project, the Japan Eye Genetic Consortium and the UK Inherited Retinal Dystrophy Consortium. (DOI: 10.1002/humu.23349),	None.

Have literature reviews on multi omics studies in rare diseases been performed and published? I believe this is the case for genomics and metabolomics, and for genomics and transcriptomics. The authors should list in the current manuscript the reviews that have been performed to date.	We have included several reviews for specific rare disease phenotypes, which highlights that our planned review will more comprehensively address and build upon previous literature reviews of multi-omics and rare disease. A PubMed search reveals only 11 reviews of multiomics and rare disease using the search terms: ((rare disease AND Review[ptyp])) AND (((multi-omics) OR multiomics) AND Review[ptyp]). However, after closer inspection it can be seen that these reviews actually highlight studies of 'single' omic analysis across the various multi-omic spectrum, rather than papers which have integrated different omic analysis, e.g. a study of transcriptomics AND genomics of a rare disease. (https://www.ncbi.nlm.nih.gov/pubmed/?term=((rare+disease+AND+Review%5Bptyp%5D))+AND+(((multi-omics)+OR+multiomics)+AND+Review%5Bptyp%5D)) In this way our review will be unique in systematically searching to identify primary studies of multiomics and rare disease.	Such review articles are now discussed briefly in the introduction section. (Lines 95 – 100)
Who do the authors include diagnostic, biomarker, and treatment studies in this review? Casting the net too wide could make the review too diffuse and difficult to interpret. Please address this concern.	We have previously performed a systematic review of methylation in rare renal disease and rare ophthalmic disorders. This earlier research supports the need for a broader review exploring multi-omic studies for rare disease(s). The authors have substantial experience in this type of analysis and are confident that we can appropriately handle the returned results. We will carefully analyse and report data for relevant components, ensuring that information is interpreted distinctly and clearly disseminated where appropriate.	None.

Most interesting for the reader would be a take away message from the literature review such as 'which multi-omics study designs are most successful? what are predictors of success? what are the do's and the don'ts?' Perhaps the authors can incorporate this into their review design.	We agree these are key outcomes from the proposed review and are keen to perform experimentally sound research, which generates results that will adequately address these questions.	We have included a short paragraph in the introduction section which reflects how we will highlight examples of good practice in any studies of multiomics and rare disease. (Lines 110 – 112)
The technology both for generating data and for interpreting data (bio informatics tools and resources) are developing at a high speed. Is it really faair to compare studies performed in 2012 to those in 2018?	We are interested to see how the multiomic analysis of rare diseases has evolved over time. We are not primarily planning to directly compare studies performed in 2012, when quality control and analytical options were very different from 2018, rather we will summarise outcomes from all relevant studies, highlight where multiple approaches have generated the same results, and describe elements of concern and best practice for future studies.	None.
Phenotyping and the extent to which this has been done and reported in the studies, will be central to success both in terms of diagnostic yield and discovery of diseases, biomarkers, treatments. How will the authors score and describe phenotyping?	Related to our response to reviewer 2's second concern, we appreciate that phenotyping information may be unclear in rare disease research studies. We will capture and report key phenotypic criteria in terms of defined diagnosis, major organ systems affected, and / or key features of a phenotype as reported by the authors of primary research papers.	We have added a column to Table S3 in our supplementary file – 'Phenotypes reported'.
Ethnicity of patients studied, sample types etc will also play a role in omics study yield; authors should list such variables in this manuscript and address inherent challenges.	We agree with the reviewer that these variables will impact results of omic studies. We intend to capture these characteristics in the participant information section of our data extraction tables, and we will highlight these differences in the discussion of our completed review. Our review will also discuss how a standardized approach to rare disease multiomic studies will improve methodological rigour of future studies.	We have now clearly stated in our methods (under 'Selection of studies, data extraction and analysis') that we will discuss variables which could affect omic analysis. (Lines 165 – 171)
Reviewer 3		

Kerr and colleagues outline their plan for a scoping review on all multiomics studies involving rare diseases. The goal of the review is to highlight the power of multiomics in understanding these rare diseases. The review is likely to achieve this goal based on what the authors plan to abstract from those studies. I have the following suggestions: 1- As the authors acknowledge as a limitation, their review methodology will miss multiomics studies that do not explicitly use the term “rare” in the title, abstract or key words. The potential misses as a result is likely substantial. May I suggest to use tools that search the full text such as Google Scholar? I understand that this comes at a cost since they will likely have to sort through many more studies that may list the word “rare” without being specifically focused on rare diseases. 2- One minor error: “two examples of which for are available”, please remove “for”.	We thank the reviewer for their feedback and are pleased that the strong likelihood of this review achieving its goal is emphasised. 1- As recommended, we have incorporated Google Scholar into our search strategy. A trial search using our PubMed search terms in Google Scholar (((multiomic*) OR multi-omic*)) AND ((rare disease* OR rare disorder OR rare cancer* OR rare syndrome*)) Sort by: Best Match, found 325 results, therefore we are confident that this search will be feasible. 2- Thank you for carefully reading the text and highlighting this typographical error, which has now been corrected.	1- Amended in methods (lines 143 and 148) and abstract section (line 36) of manuscript. 2- Typographical error corrected in methods section of manuscript
Reviewer 4		
1. Please indicate that your completed review will follow scoping review reporting guidelines from PRISMA. PRISMA-ScR extension: http://annals.org/aim/fullarticle/2700389/prisma-extension-scoping-reviews-prisma-scr-checklist-explanation	Yes, we are pleased to confirm that our completed review will follow scoping review reporting guidelines and have updated the text in the methods section to clearly reflect this.	We have now clearly indicated that our review will follow the PRISMA-ScR guidelines in methods section of the manuscript. (Lines 123 – 125)
I would recommend that your team consult a librarian or other search expert for advice on your search strategy, including which databases to search and which terms to include.	As recommended the authors have explicitly sought advice from our Faculty librarian specifically on the search strategy, including database selection and term, for this review. Additionally, one of the authors Dr Helen McAneney is a statistical editor for the Cochrane	Updated example search terms for MEDLINE and Embase via Ovid are now included in the supplementary material (Table S1 and S2)

	Developmental, Psychosocial and Learning Problems (CDPLP) review group, has also reviewed the search terms.	
It is unclear which platforms you will use to search each database. Please indicate this in your Methods--e.g., MEDLINE (Ovid), EMBASE (Embase.com), etc. There is no need to search MEDLINE via both Ovid and PubMed.	Thank you for highlighting this important point. We have now indicated which platforms we will use in the main text.	Platforms indicated in methods section. (Lines 142 – 143)
The Cochrane Database of Systematic Reviews does not identify primary studies so I'm not sure why you would include this in your search. Cochrane Reviews are indexed for MEDLINE so you would already be finding these through that database if you do want to include them as means of identifying primary studies.	We intended to include this as a comprehensive approach for identifying primary studies through review articles. However, following this expert reviewer's guidance we have removed Cochrane Database of Systematic Reviews from the list of databases we will search.	Cochrane Database of Systematic reviews removed from methods section databases to be searched. (Lines 142 – 143)
It is unnecessary to conduct the search twice. Have your search strategy reviewed by a librarian/search expert to ensure that the syntax is correct and reproducible and then run it once. On the other hand, you do need two individuals to screen articles, which you do not mention. Please indicate that two individuals will screen results, with a third resolving discrepancies.	It was our intention that the screening and data extraction be performed by independent personnel (two for primary screening with a third employed to help resolve discrepancies), which has been indicated more clearly in the text.	Indicated more clearly in the methods section. (Lines 169 – 172)
It is not clear whether your search strategy as presented is broad enough to capture all relevant literature. I understand that there exist thousands of rare diseases and so it would likely not be feasible to search for all of these by name, but it seems that you will miss a large portion of the literature by limiting to only those articles with "rare disease" or one of its	We thank the reviewer for this suggestion. However as highlighted, there are more than 8000 rare diseases, which are not practical to search for each individual condition. Thus, we have taken the pragmatic approach proposed in this protocol. As per the suggestion of reviewer 3, to address this limitation we have also included Google Scholar within our search, as this searches the full	Following consultation with the faculty librarian, our search terms have been improved and modified in tables S1 and S2 of our supplementary files. (Lines 149 – 151)

variants in the title, abstract or controlled vocabulary. Please explore whether it would be possible to include individual disease names in your search, even from an abbreviated list, in order to improve recall in your search--again, a librarian or other search expert could help you with this. Likewise "multiomics"--is there a way to identify articles that mention two individual omic terms in the abstract but do not use the specific term "multiomic"? If not possible, it is understandable but this should be explained as a limitation of your methodology.	body of the text. This will hopefully highlight more articles which may not define a specific rare disease in their title or abstract. However, the fact that rare disease meet specific EU and US definitions (5 in 10,000 or less than 200,000 people respectively), should be highlighted in the majority of articles which feature a rare disease and we hope that this will not limit our review substantially, although our main manuscript will highlight this as a limitation. We have additionally consulted with a librarian as suggested on this matter, and our search now includes examples of the most common 'omic terms' as well as the general multiomic terms.	
Scoping reviews do not include critical appraisal of methodological quality -- see p. 17 of the JBI manual that you reference. This portion of your analysis should be removed. Please consult seminal scoping review methods papers (e.g. Arksey & O'Malley 2005, Levac, Colquhoun & O'Brien 2010) in addition to JBI to ensure that you are following best practices. Consulting experts about initial findings would be a useful step to add to your method.	We thank the reviewer for highlighting this oversight, and we will remove the critical appraisal section of our methods / supplementary files. We feel it would be helpful for us to have some measure of study methodological rigour, and so this will be highlighted in the data extraction form, for example if experimental controls were used and confounding factors addressed. Additionally, participant information such as sample size, ethnicity will be captured within our data extraction form.	Methodological rigour statement removed in methods section, and a statement added that study quality will be more generally assessed using the data extraction forms (also modified in supplementary material)
Reviewer 5		
This paper presents the protocol of an open review without the objective to test a confirmatory hypothesis. So there is no risk of selective outcome reporting and I think it may be published in its actual wording. Just a minor comment. Please consider if you should also pay attention to individualised trials (i.e., different interventions in repeated periods for stable, chronic conditions, such as "n-of-1" trials).	We thank the reviewer for this feedback and are pleased to confirm that we intend to include individualised trials of multiomic analysis in rare diseases which is captured by our broad search strategy.	None.